# Reliability and Concurrent and Construct Validity of a Food Frequency Questionnaire for Pregnant Women at High Risk to Develop Fetal Growth Restriction

**DOI:** 10.3390/nu13051629

**Published:** 2021-05-12

**Authors:** Charlotte Juton, Sara Castro-Barquero, Rosa Casas, Tania Freitas, Ana Maria Ruiz-León, Francesca Crovetto, Mónica Domenech, Fátima Crispi, Eduard Vieta, Eduard Gratacós, Ramon Estruch, Helmut Schroder

**Affiliations:** 1Institut de Recerca Sant Joan de Déu, Endocrinology Department, Esplugues de Llobregat, 08950 Barcelona, Spain; charlottejuton@gmail.com; 2PhD Program, Food and Nutrition University of Barcelona, 08007 Barcelona, Spain; 3Department of Internal Medicine Hospital Clinic, Institut d’Investigacions Biomèdiques August Pi i Sunyer (IDIBAPS), University of Barcelona, 08007 Barcelona, Spain; sacastro@clinic.cat (S.C.-B.); FREITA@clinic.cat (T.F.); amruiz@clinic.cat (A.M.R.-L.); MDOMEN@clinic.cat (M.D.); restruch@clinic.cat (R.E.); 4Centro de Investigación Biomédica en Red de Fisiopatología de la Obesidad y Nutrición (CIBEROBN), ISCIII, 28029 Madrid, Spain; 5BCNatal | Fetal Medicine Research Centre, Hospital Clínic and Hospital Sant Joan de Déu, University of Barcelona, IDIBAPS, 08007 Barcelona, Spain; FCROVETTO@clinic.cat (F.C.); FCRISPI@clinic.cat (F.C.); GRATACOS@clinic.cat (E.G.); 6Centro de Investigación Biomédica en Red de Enfermedades Raras (CIBERER), Instituto de Salud Carlos III, 28029 Madrid, Spain; 7Departments of Psychiatry and Psychology, Hospital Clinic, Neuroscience Institute, IDIBAPS, University of Barcelona, 08007 Barcelona, Spain; EVIETA@clinic.cat; 8Centro de Investigación Biomédica en Red en Salud Mental (CIBERSAM), ISCIII, 28029 Madrid, Spain; 9Cardiovascular Epidemiology and Nutrition Research Group, Institut Hospital del Mar d’Investigacions Biomèdiques (IMIM), 08003 Barcelona, Spain; 10Centro de Investigación Biomédica en Red de Epidemiología y Salud Pública (CIBERESP), Instituto de Salud Carlos III (ISCIII), 28220 Madrid, Spain

**Keywords:** food frequency questionnaire, validity, reliability, Mediterranean diet, pregnancy, fetal growth restriction

## Abstract

Accuracy of dietary assessment instruments such as food frequency questionnaire (FFQ) is crucial in the evaluation of diet–disease relationships. Test–retest reliability and concurrent and construct validity of a FFQ were evaluated in 150 pregnant women at high risk to develop fetal growth restriction randomly selected from those included in the improving mothers for better prenatal care trial Barcelona (IMPACT BCN). The FFQ and dietary records were performed at baseline and 34–36 weeks of gestation. Test–retest reliability of the FFQ for 12 food groups and 17 nutrients was moderate (ICC = 0.55) and good (ICC = 0.60), respectively. Concurrent validity between food, nutrients and a composite Mediterranean diet score (MedDiet score) and food records was fair for foods and nutrients (ρ average = 0.38 and 0.32, respectively) and moderate (r = 0.46) for the MedDiet score. Validation with biological markers ranged from poor (r = 0.07) for olives to moderate (r = 0.41) for nuts. A fair concordance between methods were found for nutrients (weighted κ = 0.22) and foods (weighted κ = 0.27). The FFQ-derived MedDiet score correlated in anticipated directions with intakes of nutrients and foods derived by food records. The FFQ showed a moderate test–retest reliability and reasonable validity to rank women according to their food and nutrient consumption and adherence to the Mediterranean diet.

## 1. Introduction

Monitoring the diet quality of pregnant women is crucial for the improved health of both themselves and their child. Good nutrition during pregnancy may help to prevent deficiencies that affect the health of both the mother and fetus [1,2]. Women following a Mediterranean diet (MedDiet) during pregnancy have displayed superior health for both themselves and their offspring [3]. The economic benefits of using food frequency questionnaires (FFQs), compared to more expensive methods such as 24 h recalls (24 hR) or food records [4], make them a popular tool to estimate food intake for different populations. However, the validity of an FFQ is limited to the target population in which it was validated. Thus, an FFQ validated in the general population [5] should not be administered to pregnant women.

The accuracy of an FFQ can be determined by comparing food and nutrient intake information collected by this instrument with those obtained using a reference method. Weighed food records are considered the gold standard method; however, the application of this method is often not feasible. Therefore, food records and 24 hR are widely used as the reference method. However, these instruments are not free from measurement errors caused by memory bias among others [4]. These markers are in contrast to auto reported data objective measures of food intake. It is, therefore, appropriate to include additional biological markers of food intake in the evaluation of the validity of an FFQ [6].

Most FFQ validation studies compare intake estimates collected by an FFQ with those collected by a reference method to assess their agreement in terms of reported intakes of individual nutrients and/or food groups. However, it should be acknowledged that these results cannot yield a generalized score composed of several specific nutrients and/or foods like the Mediterranean diet score (MedDiet score).

Given the great impact of diet quality on health [1,3,7], it is important that FFQs correctly estimate food and nutrient intake. However, most FFQ validation studies have not gone further than analyzing the presentation of data on test–retest reliability and concurrent validity of the questionnaire. Thus, this study aimed to determine the test–retest reliability indicating the stability of the FFQ over time through repeated measures at two different time points. Furthermore, we determined concurrent or relative validity, which indicates the amount of agreement between two different measures, and construct validity a measure of the concept that it is intended to measure. For this purpose, we compared dietary data obtained by the FFQ with those derived by food records and biological markers of food intake. Additionally, we analyzed these validity domains along with changes in adherence to the MedDiet.

## 2. Methods

### 2.1. Study Population

The present validation study was performed for the FFQ for women participating in improving mothers for better prenatal care trial Barcelona (IMPACT BCN) (NCT03846206). Eligible participants for the IMPACT trial were pregnant women with a high risk of developing fetal growth restriction (FGR) during pregnancy (odds ratio, OR >2), according to the criteria of the Royal College of Obstetricians and Gynecologists (RCOG) [8]. We randomly selected women attending their second-semester scan (from 19–23 weeks of gestation) to participate. Exclusion criteria were any of the following: fetal anomalies including chromosomal abnormalities, structural malformations or congenital infections detected prenatally; neonatal abnormalities diagnosed after birth; no possibility to perform additional visits; participation in another trial and maternal mental retardation or other mental or psychiatric disorders. After providing informed consent, they were randomized into three equally-sized intervention groups: (1) nutrition program based on MedDiet; (2) stress-reduction program based on mindfulness techniques and (3) control group with no specific intervention (usual care). From a total of 304 pregnant women who agreed to participate, we randomly selected 150 (50 per group). Complete data on plasma concentrations of vitamin B12, folic acid and linolenic acid and urine hydroxyl tyrosol were only available for 109 of the participants.

The study was approved by the Institutional Review Ethics Committee of the Hospital Clinic, Barcelona (HCB-2016-0830) and registered with the ClinicalTrials.gov identifier number NCT03166332. The study protocol was described elsewhere [9]. Signed informed consent was obtained for all participants.

### 2.2. Dietary Assessment

Dietary assessment was self-reported by the participants and performed at baseline (19–24 weeks) and at the final visit (34–36 weeks of gestation). The participants were asked about their dietary habits during the last year. Food consumption was estimated by a slightly modified and validated semiquantitative optical readable FFQ [5] administered by a trained interviewer. In a 151-item food list including alcoholic and non-alcoholic beverages (typical foods in Spain), participants indicated their usual consumption and chose from nine frequency categories, ranging from never or <1 time/month to ≥6 times/day. Food items were listed under 14 food groups: milk and dairy products, cereals and whole grains, vegetables, legumes, sausages, oils and fats, eggs, meat and fish, fast food, canned products, fruit, nuts, sweets and desserts and others (salt and sugar) and alcoholic and non-alcoholic beverages. The original questionnaire had 137 items. We added few food-items to our FFQ in order to adapt the habitual food products consumption in our geographical area, such as sweeteners, soy milk and other plant-based milks, avocado, dark chocolate, different snacks and non-alcoholic beverages.

Seven-day food records were collected at baseline and follow-up final visits. The participants filled a 7-day food diary of the previous 7 days before the meeting. Detailed information about their dietary intake, including the estimated grams or portions sizes was collected. Moreover, in the food diary the instructions with the portion sizes and how to provide this information in the food dairy was also described. Three days out of the seven-day food records, including two working days and one day in the weekend, were collected during each visit. Food consumption derived by the FFQ and food records was converted into energy and nutrient intake by the CESNID and Moreiras composition tables using traditional recipes [10,11].

### 2.3. Calculation of the Mediterranean Diet Score (MedDiet Score)

The MedDiet score was calculated according to the Trichopoulou and colleagues’ method at baseline and follow-up [12]. The MedDiet score calculated at follow-up was used to assess concurrent validity. Additionally, biological markers analyzed at follow-up were used for the correlation with the corresponding food intake at follow-up. Median distribution of cereals, dairy products, fish, meat, legumes, vegetables, fruits and nuts, and the ratio between monounsaturated to saturated fatty acids (MUFA/SFA) were calculated. One point was assigned for intake of each food group of cereals, fruits and nuts, vegetables, legumes and fish and for MUFA/SFA equal or above the median and zero points for the consumption below the median. The consumption of dairy products and meat was reverse-coded. The final score ranged from 0 to 8 units. Alcohol intake was not included in the calculation because, for the participants who identified themselves as alcohol consumers (32.7%), the average intake of alcohol was below 1 g a day.

### 2.4. Biological Markers

Routine analyses were performed at the CORE laboratory of the Hospital Clinic of Barcelona, which fulfilled all the required quality criteria for the study. Blood samples were drawn in the morning (after a minimum of 10 h of fasting) for B12 vitamin and folic acid analyses. B12 vitamin and serum folic acid were measured by automated electrochemiluminescence immunoassay system Advia-Centaur, Siemens (Siemens Tarrytown, NJ, USA), with reagents provided by the instrument manufacturer. The estimation of hydroxytyrosol and its glucuronides levels were measured following the method proposed by Khymenets et al. [13] in the Department of Nutrition, Food Sciences and Gastronomy, XaRTA, School of Pharmacy and Food Sciences, University of Barcelona. Briefly, 0.5 mL of urine was acidified with H3PO4 at 4% and then extracted with solid-phase extraction (SPE) using Oasis^®^ HLB 3 cc (60 mg) cartridges (Waters Corporation, Ireland). The extract was evaporated under an N2 curtain and reconstituted in 200 uL of 1 mM ammonium acetate at pH 5. The identification and quantification of these compounds were analyzed with UPLC-MS/MS analyses.

Plasma α-linolenic acid concentrations were assessed by gas chromatography [14]. The analysis was performed on a Shimadzu GC-2010 gas chromatograph equipped with a flame ionization detector (Shimadzu, Kyoto, Japan), with capillary columns of 10 m × 0.10 mm × 0.10 μm film thickness (Varian, Palo Alto, CA, USA). Determinations of individual fatty acids were estimated according to their retention time during gas chromatography and were quantified as a percentage by weight (% by weight) of total plasma fatty acids.

The limits of quantification for biological markers were 9.86–32.89 ng/mL, 10.0–16.8 ng/mL, equal or less than 90 pg/mL and equal or less than 0.70 ng/mL for hydroxytyrosol, fatty acids, vitamin B12 and folic acid, respectively.

We chose these biomarkers because they represent an objective measure of typical Mediterranean foods such as olive oil, olives nuts and fish.

### 2.5. Other Variables

Height and weight were measured at baseline and final visit. Information on demographic and socioeconomic variables and tobacco smoking were obtained through structured standard questionnaires administered by trained personnel. Socioeconomic status was defined as low if the subject had either never been in employment or had been unemployed for more than two consecutive years, medium if the subject had completed secondary studies and was currently in employment and high if the subject had completed university studies and was currently in employment.

### 2.6. Statistical Analysis

Mean with standard deviation (SD) and proportions (%) were calculated for the characteristics of the subjects (shown in Table 1). Intraclass correlation coefficients between basal and follow-up data on FFQ-derived nutrients and food categories were determined for test–retest reliability. For the analysis of concurrent (relative) and construct validity we used data from the FFQ administer at follow-up and data from food records (reference method). The relative validity of the FFQ (test method) against the food records (reference method) was first assessed by calculating the Spearman correlation coefficient. Additionally, we determined Pearson correlation coefficients between biological markers and the corresponding food and nutrient intake. However, it should be observed that two highly correlated measures can still show considerable differences between the two measurements across their range of values. We thus calculated the absolute agreement of categorical variables between the two measurements by cross-classification and κ statistic of tertile distribution of food categories and nutrients. Values of κ were organized as follows: >0.8 for almost perfect agreement, between 0.61 and 0.80 for substantial agreement, 0.41 and 0.60 for moderate agreement, 0.21 and 0.40 for fair agreement and ≤0.20 for slight agreement [15]. The concurrent validity and absolute agreement of the MedDiet adherence scores derived by the FFQ (test method) and the dietary records (reference method) were also tested. Agreement between the scores obtained by the FFQ and dietary records was then assessed using the Bland–Altman method [16] and the intraclass correlation coefficient (ICC). These methods assess the agreement between two methods by calculating the mean of their differences and regressing that figure against the average score obtained by the two methods. A complete agreement between the methods would involve a mean proportional agreement of 100% and a mean difference of 0 between the scores derived by both measurements. Proportional bias represented by possible variations in the level of agreement between methods was also analyzed. We then fitted linear regression models, with the mean instrument differences of the FFQ- and dietary records-derived MedDiet scores (FFQ–dietary records) constituting the dependent variable and the mean score of both ((FFQ + dietary records)/2)) constituting the independent variable. Finally, to assess construct validity, general linear modelling was used to estimate associations between energy-adjusted nutrient intakes derived from food records and Mediterranean diet adherence (tertile distribution) calculated from the FFQ. Linear trends were tested by including the categorized variable (tertile distribution of the scores) as continuous in this model. The polynomial contrast was used to determine *p* for the linear trends for continuous variables and a post hoc Bonferroni correction for multiple comparisons was conducted. The Statistical Package for the Social Sciences statistical software package version 21.0 (SPSS Inc., Chicago, IL, USA) was used for all statistical analyses. Differences were considered significant if *p* was <0.05.

## 3. Results

General characteristics of the study population are shown in Table 1. The mean of the MedDiet score at baseline and follow up was 4.0 ± 1.5 and 4.1 ± 1.6, respectively. Table 2 shows the test–retest reliability of the FFQ administered at 19–24 and 34–36 weeks of gestation. We found a significant test–retest reliability for each of the 12 food groups and 17 nutrients. The ICC ranged from moderate (cereal = 0.42) to very good (vegetables = 0.83) for foods and from moderate (MUFA = 0.48) to very good (Vitamin A = 0.86) for nutrients. The average correlation of foods and nutrients was 0.70 and 0.69, respectively. Test–retest reliability for the MedDiet score showed an ICC of 0.53, considerably lower than for foods and nutrients.

The analysis of concurrent validity (Table 3) yielded correlations ranging from poor to good for foods and nutrients. The average correlations of foods and nutrients between the two methods were 0.24 and 0.21, respectively. All correlations were significant with the exception for olive oil and vitamin B12 for non-standardized and olive oil, vitamin B12 and monounsaturated fat for standardized foods and nutrients (Table 3 and Table 4). The degree of correlations ranged from poor (olive oil ρ = 0.12) to good (dairy products ρ = 0.63) for foods and from poor (vitamin B12 ρ = 0.10) to moderate (vitamin C ρ = 0.47) for nutrients (Table 3). Concordance between methods ranged from poor to fair for foods and nutrients. κ statistic showed a fair concordance between the methods for foods (weighted κ = 0.28) and nutrients (weighted κ = 0.21). The adjustment for energy intake did not meaningfully improve correlations or concordance between the methods (Table 4). The comparison of the biomarker α-linolenic acid was limited to nut consumption because there were no other relevant food sources α-linolenic acid included in the FFQ. Hydroxytyrosol was correlated with olive oil and olives, both principal sources of this bioactive compound. The comparison between biomarkers of food and nutrient intake with the corresponding data derived by the FFQ revealed a poor (r = 0.07; *p* = 0.59) to moderate (r = 0.41; *p* <0.001) concurrent validity for hydroxytyrosol and α-linolenic acid, respectively (Table 5).

The mean ratings of the MedDiet score derived by the FFQ at baseline and follow-up 4.0 ± 1.5 and 4.1 ± 1.6, respectively, and for dietary records 4.0 ± 1.5. The FFQ significantly (*p* < 0.001) overestimated the MedDiet score (by 12%) compared to the corresponding MedDiet score derived by the reference method. However, no proportional bias was found (β coefficient 0.072; 95% CI-0.064, 0.204; *p* < 0.287) (Table 6 and Figure 1) across score ratings. The Pearson coefficient revealed a moderate and significant correlation (0.46 and *p* < 0.001) between the scores derived by the dietary records and FFQ. Additionally, the intraclass correlation coefficient, an indicator of the degree to which both instruments assigned the same absolute score ratings, showed the same degree of correlation (ICC = 0.46; *p* < 0.001). These findings indicate that the FFQ had a moderate ability to rank participants according to their adherence to the MedDiet. To analyze construct validity, we hypothesized a priori relationships between higher scores of the more favorable intake profiles for 17 nutrients. We found that intakes of 17 nutrients were associated in the anticipated direction with MedDiet score ratings derived by the FFQ, the associations were significant for nearly 50% of the nutrients (Appendix A).

## 4. Discussion

The results of this validation study show a good test–retest reliability and a fair to moderate validity of the FFQ for pregnant women. The questionnaire adequately ranked women according to their adherence to the MedDiet. Additionally, the construct of the MedDiet score was valid.

FFQs are useful tools for assessing long-term dietary intake in epidemiological studies [17]. Few FFQs capturing the complete diet have been developed and validated in European populations of pregnant women during the last 20 years [18,19,20,21,22,23,24]. Of those that have, only test–retest reliability and concurrent validity were analyzed, using 24 hRs or dietary records as the reference method. In this study, the average test–retest reliability for 12 food groups and 17 nutrients was moderate (ICC = 0.55) and good (ICC = 0.61), respectively. This is somewhat higher than that found by Vioque et al. [21] who reported moderate reliability for 29 nutrients (r = 0.51) and 17 foods (r = 0.41). A Finnish study found a higher average correlation (ICC = 0.65) for all foods and nutrients [18]. However, these comparisons are somewhat limited due to the different amounts of foods and nutrients considered and, in the case of Vioque et al. [21], different statistical methods used. Additionally, the test–retest reliability of diet in these studies is based on the assumption that differences found between the two estimations are mainly due to measurement errors and less to alterations in dietary habits. In the present study one third of the pregnant women were allocated to a nutritional intervention program and have, therefore, changed their dietary habits during the test–retest period. This fact might partially explain the magnitude of the test–retest reliability of the present FFQ

In this study we found a poor (olive oil ρ = 0.12) to good (dairy products ρ = 0.63) concurrent validity of the FFQ for foods and from poor (vitamin B12 ρ = 0.10) to moderate (vitamin C ρ = 0.47) for nutrients compared with dietary records These findings are comparable with previous reports of the validity of FFQs in European pregnant women [22,23,24,25,26,27]. The poor concurrent validity of olive oil in the present study was somewhat surprising because it is a characteristic food of the Mediterranean diet and hence it is unlikely that the relatively short reporting time of the reference method was a reason for this finding. However, the objective measure of olive oil consumption by its corresponding biological marker revealed a better correlation. In contrast, a poor correlation of vitamin B12 derived by the FFQ with the corresponding data from the reference method and the biological marker were found for both. One might argue that this possibly reflects the difficulty in estimating meat servings but the correlations of meat and processed meat between methods were substantially better than that for vitamin B12. The poor correlation of vitamin B12 derived by the FFQ with its biological marker might be biased by dietary supplements containing vitamin B12.

Drawing a fair comparison between the concordance between the methods applied in this study and that of other publications presents a challenge, as most validation studies present proportional agreement instead of kappa statistics. The proportional agreement is easily understandable although it does not consider coincidental occurrences of agreement between two different measures. In this study, the concordance between food and nutrient intake derived by the FFQ and food records ranged from poor to moderate with an average weighted kappa of 0.21 and 0.28 for foods and nutrients, respectively. This finding indicates a fair overall concordance between the FFQ and reference methods.

Food records and 24 hRs are frequently used as gold standards in validation studies of dietary assessment [17]. These reference methods are not free from measurement errors, however, the type of error is independent of those from an FFQ [17]. The objective measurement of food and nutrient intake by corresponding biological markers yielded a more robust estimate of an FFQs validity. In this study, objective measurements of plasma levels of both folic acid and linolenic acid and urinary hydroxyltyrosol were fairly correlated (*p* < 0.05) with their corresponding FFQ-derived nutrient or food. Poor correlations were found for olives (r = 0.07; *p* = 0.58) and vitamin B12 (r = 0.09; *p* = 0.30). The magnitude of correlation was somewhat better for folic acid (0.12 vs. 0.25) but similar for vitamin B12 than that found by Vioque et al. [21]. Usually, validation studies compare intake estimates between the collected data from the desired instrument against a reference method to assess whether it correctly classifies reported intakes of nutrients and food groups. From these results, nothing can be deduced regarding the accuracy of the ranking for a composite score of multiple nutrients and/or foods like the MedDiet score; evidence for the validity of predefined indices, such as the MedDiet score, is scarce [22,23,24]. Benitez-Arciniega et al. [25] also reported a moderate (r = 0.48) correlation between a modified MedDiet score derived by an FFQ and repeated 24 h dietary recalls in a Mediterranean population. Stronger concurrent validity was found for the traditional and alternate MedDiet score in German women [27] and a multiethnic Asian population, respectively [27]. In comparison with this study, the agreement between test and reference method was considerably stronger in the Asian group but only slightly different in the German population.

Construct validity should also be considered when selecting a dietary assessment tool. We hypothesized that both the FFQ- and dietary records-derived dietary quality scores would show a positive correlation and that both of the FFQ-derived dietary quality indices would be positively associated with a favorable nutrient intake profile estimated by food records. Intakes of 17 nutrients were significantly associated in the anticipated direction with MedDiet score ratings derived by the FFQ. These findings are in line with that of Benitez-Arciniega et al. [25] who reported good construct validity of two MedDiet scores by correlating nutrient intake derived by multiple food records with the MedDiet scores.

The strength of the present study is that validity was determined by correlations with foods and nutrients derived by a self-reported reference method and biological markers of food and nutrient intake. An additional strength is the inclusion of the validity of MedDiet adherence by a composite score. Finally, as in all validation studies, an inherent limitation is that reference methods such as multiple dietary recalls or records are themselves not free from error [17]. Food records for example requires motivated subjects and place a high burden on the participants. The ideal choice of the reference method is weighed food records, which is considered the “gold standard”. However, the administration of food records or weighed food records may lead to participants changing their diet during a recording period. Food frequency questionnaires on the other hand are prone to memory bias because these questionnaires asked for the retrospective food intake. Furthermore, average consumption frequency of seasonal foods is especially critical and the fixed food list in fixed portion sizes are other sources of measurement error. Finally, the use of the FFQ to present data of absolute intakes of foods and nutrients is limited without prior calibration of these data by a reference method. This is especially the case for foods and nutrients with poor concurrent validity and concordance.

## 5. Conclusions

In conclusion, the present FFQ is a dietary assessment instrument with reasonable validity in its application to a population of pregnant women.

## Figures and Tables

**Figure 1 nutrients-13-01629-f001:**
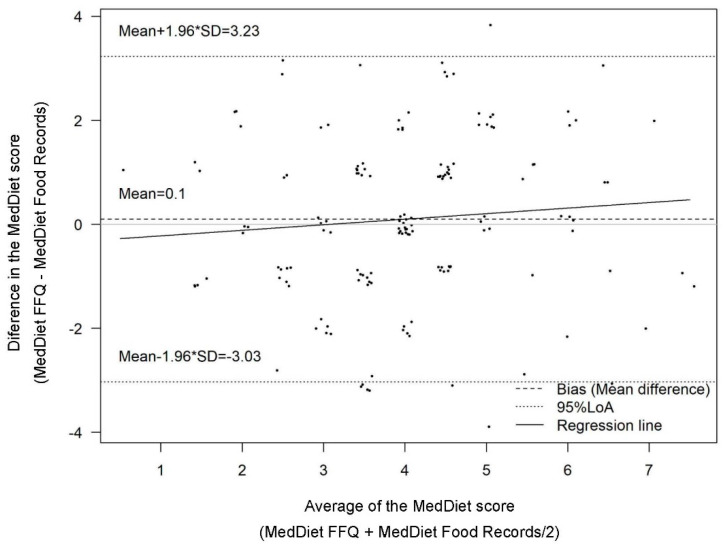
Bland-Altman plot for the agreement of the MedDiet score derived from the food frequency questionnaire and the reference method (*n* = 150).

**Table 1 nutrients-13-01629-t001:** General characteristics of the participants (*n* = 150).

Variables	
Age, years (SD)	38.0 (4.0)
Weight, kg (SD)	65.8 (12.8)
Height, cm (SD)	162 (6)
Education, %	
- High	69.3
- Medium	26.7
- Low	4
Current smokers, %	6
Alcohol consumer, %	32.7 *
Energy intake, kcal/d	2412 ± 500

* Below a mean of 1 g a day.

**Table 2 nutrients-13-01629-t002:** Test–retest reliability of the food frequency questionnaire (*n* = 150).

Food Groups ^1^ (g)	ICC ^2^	95% CI
Cereals	0.26	0.11–0.41
Legumes	0.55	0.42–0.65
Vegetables	0.71	0.62–0.78
Fruits	0.53	0.41–0.64
Nuts	0.51	0.37–0.63
Dairy products	0.67	0.57–0.75
Fish	0.62	0.51–0.71
Meat	0.58	0.46–0.68
Processed meat	0.63	0.52–0.72
Olive oil	0.54	0.41–0.64
Animal fat	0.62	0.43–0.76
Baked goods	0.36	0.21–0.50
**Nutrients ^1^ (units)**	ICC	95% CI
Energy (kcal)	0.57	0.45–0.67
Carbohydrates (g)	0.58	0.46–0.68
Proteins (g)	0.62	0.51–0.71
Fat (g)	0.50	0.54–0.76
Saturated FA (g)	0.51	0.38–0.77
Monounsaturated FA (g)	0.48	0.35–0.59
Polyunsaturated FA (g)	0.49	0.36–0.60
Cholesterol (mg)	0.66	0.55–0.74
Fiber (g)	0.68	0.59–0.76
Potassium (mg)	0.67	0.57–0.75
Calcium (mg)	0.61	0.50–0.70
Vitamin C (mg)	0.62	0.51–0.71
Vitamin A (mcg)	0.76	0.68–0.82
Vitamin E (mg)	0.59	0.47–0.68
Niacin (mg)	0.62	0.51–0.71
Folic acid (mcg)	0.70	0.60–0.77
Vitamin B_12_ (mcg)	0.68	0.59–0.76
**Score (units)**	ICC	95% CI
MedDiet score	0.53	0.40–0.63

ICC = intraclass correlation coefficient; CI = confidence interval; FA = fatty acids; MedDiet = Mediterranean diet.^1^ Variables were log-transformed with the exception of energy and the MedDiet score.^2^ *p* < 0.01 for all.

**Table 3 nutrients-13-01629-t003:** Concurrent validity and concordance of foods and nutrients derived by the food frequency questionnaire and dietary records (*n* = 150).

Food Groups (g/day)	FFQ			24 h Recalls			Spearman’s Correlation Coefficient ^1^	Magnitude of Association	Weighted κ
	Median	P25	P75	Median	P25	P75			
Cereals	94.3	77.1	113	146	119	180	0.29	Fair	0.23
Legumes	52.9	37.4	72.9	22.5	3.14	41.1	0.37	Fair	0.26
Vegetables	351	258	448	180	130	245	0.26	Fair	0.13
Fruits	374	255	501	204	137	283	0.53	Moderate	0.39
Nuts	17.1	6.0	32.0	4.33	0	10.0	0.46	Moderate	0.30
Dairy products	337	230	527	245	159	336	0.63	Good	0.42
Fish	78.8	53.1	108	60.0	34.6	80.0	0.40	Fair	0.29
Meat	107	74.3	149	60.0	35.0	91.7	0.30	Fair	0.24
Processed meat	40.6	21.4	60.1	40.4	22.3	59.2	0.38	Fair	0.29
Olive oil	50.0	50.0	50.0	35.0	30.9	35.9	0.12	Poor	-
Animal fat	0	0	5.14	0	0	3.35	0.47	Moderate	0.43
Baked goods	30.5	14.3	50.3	52.1	34.8	86.8	0.34	Fair	0.14
**Nutrients (units/day)**									
Energy (kcal)	2447	2133	2702	2001	1759	2198	0.29	Fair	0.16
Carbohydrates (g)	207	173	255	205	180	239	0.27	Fair	0.14
Proteins (g)	109	89.1	127	87.6	77.1	98.5	0.41	Moderate	0.28
Fat (g)	126	110	144	93.2	81.5	104	0.33	Fair	0.25
Saturated FA (g)	33.1	27.1	39.9	26.2	21.7	30.6	0.29	Fair	0.17
Monounsaturated FA (g)	62.2	53.5	71.5	42.3	38.6	47.0	0.28	Fair	0.17
Polyunsaturated FA (g)	20.8	16.4	27.5	13.8	11.2	17.2	0.37	Fair	0.25
Fibers (g)	34.0	26.9	42.0	21.8	18.6	25.3	0.42	Moderate	0.25
Cholesterol (mg)	313	270	395	294	240	350	0.26	Fair	0.17
Potassium (mg/day)	4530	3916	5517	2664	2213	3059	0.38	Fair	0.26
Calcium (mg)	1089	834	1371	869	727	1043	0.44	Moderate	0.30
Vitamin C (mg)	259	187	344	117	86.4	158	0.47	Moderate	0.29
Vitamin A (mcg)	1354	943	1754	1072	690	1433	0.28	Fair	0.16
Vitamin E (mg)	18.2	15.1	22.0	9.1	7.99	10.5	0.34	Fair	0.25
Niacin (mg)	24.9	21.2	29.1	52.5	45.3	61.3	0.27	Fair	0.19
Folic acid (mcg)	495	391	599	371	292	431	0.26	Fair	0.22
Vitamin B_12_ (mcg)	7.1	5.40	10.4	5.15	3.66	8.33	0.11	Poor	0.10

^1^ *p* < 0.05 for all with the exception of olive oil and vitamin B12.

**Table 4 nutrients-13-01629-t004:** Concurrent validity and concordance of foods and nutrients per 1000 kcal derived by the food frequency questionnaire and 24 h recalls (*n* = 150).

Food Groups (g/100 kcal/day)	FFQ			24 h Recalls			Spearman’s Correlation Coefficient ^1^	Magnitude of Association	Weighted κ
	**Median**	**P25**	**P75**	**Median**	**P25**	**P75**			
Cereals	39.5	33.2	47.7	73.0	59.2	87.2	0.25	Fair	0.19
Legumes	22.2	15.5	31.3	11.6	1.46	20.7	0.33	Fair	0.18
Vegetables	143	111	179	93.8	67.2	120	0.33	Fair	0.23
Fruits	149	106	200	103	71.8	141	0.50	Moderate	0.28
Nuts	7.71	2.83	12.7	2.18	0	5.0	0.47	Moderate	0.34
Dairy products	135	94.7	209	127	85.2	170	0.65	Good	0.45
Fish	33.8	21.9	45.9	28.8	17.3	41.3	0.43	Moderate	0.26
Meat	43.3	31.7	56.5	31.8	17.6	45.0	0.29	Fair	0.18
Processed meat	16.8	10.0	25.0	20.0	10.8	28.4	0.40	Fair	0.23
Olive oil	19.2	15.3	22.3	16.6	14.9	18.8	0.07	Poor	0.04
Animal fat	0	0	1.40	0	0	1.81	0.47	Moderate	0.42
Baked goods	12.5	6.11	20.1	27.8	16.5	43.1	0.36	Fair	0.20
**Nutrients (units/day)**									
Carbohydrates (g)	87.1	79.1	94.8	103	96.2	112	0.42	Moderate	0.27
Proteins (g)	44.1	39.2	50.4	43.3	38.9	48.9	0.58	Moderate	0.37
Fat (g)	52.2	48.2	56.2	47.0	43.9	49.7	0.25	Fair	0.14
Saturated FA (g)	13.7	12.3	15.3	13.3	11.4	14.7	0.39	Fair	0.26
Monounsaturated FA (g)	25.4	23.2	27.6	21.7	19.5	22.8	0.12	Poor	0.11
Polyunsaturated FA (g)	8.7	7.33	10.6	6.86	5.92	8.02	0.44	Moderate	0.32
Fibres (g)	14.2	12.0	16.3	10.9	9.31	12.7	0.37	Fair	0.23
Cholesterol (mg)	133	117	155	149	123	175	0.31	Fair	0.16
Potassium (mg/day)	1901	1667	2092	1314	1179	1544	0.39	Fair	0.26
Calcium (mg)	446	360	538	436	362	520	0.57	Moderate	0.40
Vitamin C (mg)	109	77.3	132	59.9	42.5	82.6	0.46	Moderate	0.26
Vitamin A (mcg)	538	415	699	550	341	752	0.28	Fair	0.23
Vitamin E (mg)	7.48	6.58	8.57	4.62	4.07	5.12	0.25	Fair	0.21
Niacin (mg/day)	10.2	9.02	11.4	27.1	22.6	30.8	0.31	Fair	0.18
Folic acid (mcg)	207	178	236	179	150	218	0.20	Poor	0.16
Vitamin B_12_ (mcg)	2.96	2.33	4.08	2.52	1.94	4.15	0.08	Poor	0.07

^1^ *p* < 0.05 with the exception of olive oil, vitamin B12 and monounsaturated fat.

**Table 5 nutrients-13-01629-t005:** Pearson correlations (r) between biological markers and related nutrient or food intake derived by the food frequency questionnaire (FFQ) (*n* = 109).

FFQ	Biological Marker	r	*p*-Value	r Adjusted ^1^	*p*-Value
Nuts	α-linolenic acid	0.41	<0.001	0.23	0.010
Olive oil	Hydroxytyrosol	0.23	0.015	0.23	0.004
Olives	Hydroxytyrosol	0.07	0.587	0.05	0.615
Folic acid	Folic acid	0.25	0.010	0.24	0.012
Vitamin B_12_	Vitamin B_12_	0.09	0.333	0.10	0.304

All variables were log-transformed before analysis. ^1^ Standardized to 1000 kcal.

**Table 6 nutrients-13-01629-t006:** Correlation coefficients and between-method agreement of the Mediterranean diet adherence score.

	n = 150
Mean FFQ, unit (SD)	4.1 (1.6)
Mean dietary records, unit (SD)	4.0 (1.5)
Between-method difference, unit (SD) ^1^	0.1 (0)
Proportional agreement, % (95% CI) ^2^	113 (109, 124)
Upper LOA	3.2
Lower LOA	−3.0
Regression coefficient,^3^ (95% CI)	0.072 (−0.061, 0.204)
Pearson correlation coefficient	0.46
Intra-class correlation coefficient	0.46
Absolute agreement, % ^4^	57
Gross misclassification, % ^5^	11
κ ^6^	0.28

CI = confidence interval, FFQ = food frequency questionnaire, LOA = limits of agreement, MedDiet score = Mediterranean diet score. ^1^ Calculated as FFQ MedDiet score—dietary records MedDiet score. ^2^ Calculated as (FFQ MedDiet score/dietary records MedDiet score) × 100. ^3^ Regression coefficients (β) between the mean of the MedDiet score of both methods and the mean difference between both methods (independent variable). ^4^ Correctly classified tertiles of the MedDiet score derived by the FFQ and dietary records. ^5^ Opposite tertiles of the MedDiet score derived by the FFQ and dietary records. ^6^ Weighted κ between tertiles of the MedDiet score derived by the FFQ and dietary records.

## Data Availability

Data sharing not applicable. No new data were created or analyzed in this study. Data sharing is not applicable to this article.

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
