# Peer review of "Reliability and Concurrent and Construct Validity of a Food Frequency Questionnaire for Pregnant Women at High Risk to Develop Fetal Growth Restriction"

_nutrients, 2021, doi:10.3390/nu13051629_

Round 1

Reviewer 1 Report

Line 63- Please re-write the sentences “health of both the mother and fetus, whereas poor prenatal nutrition can impair children’s neurological development and health”, impaired neurological development is included in health problems.

Line 76 – Where is written “It is, therefore, appropriate to include additional biological markers of food intake in the evaluation of the validity of an FFQ”, it does not turn clear how the inclusion of biological markers will improve the validity of an FFQ, I believe as compared with gold standard methods. Please explain it better. The chosen parameters should also be justified.

Line 86 – The meaning of the term’s reliability, construct and concurrent validity could be briefly explained in the introduction, as may be reader dealing with questionnaires validation for the first time.

Line 89 – The author mention “we analyzed these validity domains along with changes in adherence to the MedDiet”. Why? Why specifically the MedDiet?

Line 135 – Minimum validation parameters of the analytical methods should be presented, namely the limits of quantification.

Line 157 – Where smoking pregnant women included in the analysis?

Exclusion criteria of participants are not described.

Line 166 – The authors should include the methodology used to control covariables.

Tables 1 and 6 – In the first 3 lines there is no indication that mean values are being presented, only SD.

Tables 2, 3 and 4 – p values of Interclass or Spearman correlations should be presented, and more considered while discussing the results. Why only Table 5?

Line 281 – The sentence “The MDS (p <0.001) significantly overestimated the 281 ratings (by 12%) compared to the corresponding score obtained by the FFQ” is not clear.

Author Response

Line 63- Please re-write the sentences “health of both the mother and fetus, whereas poor prenatal nutrition can impair children’s neurological development and health”, impaired neurological development is included in health problems.

Reply: This has been done. The sentence reads now: Good nutrition during pregnancy may help to prevent deficiencies that affect the health of both the mother and fetus [1] [2].“ (lines 65,66)

Line 76 – Where is written “It is, therefore, appropriate to include additional biological markers of food intake in the evaluation of the validity of an FFQ”, it does not turn clear how the inclusion of biological markers will improve the validity of an FFQ, I believe as compared with gold standard methods. Please explain it better. The chosen parameters should also be justified.

Reply: We have modified the corresponding paragraph for more clarity as follows:

“The accuracy of an FFQ can be determined by comparing food and nutrient intake information collected by this instrument with those obtained using a reference method. Weighed food records are considered the gold standard method; however, the application of this method is often not feasible. Therefore, food records and 24hR are widely used as the reference method. But these instruments are not free from measurement errors caused by memory bias among others [4]. These markers are in contrast to auto reported data objective measures of food intake. It is, therefore, appropriate to include additional biological markers of food intake in the evaluation of the validity of an FFQ.” (lines: 75-81)

We chose these biomarkers because they represent an objective measure of typical Mediterranean foods such as olive oil, olives, nuts, and fish. We have inserted this sentence in the method section (line: 218, 219)

Line 86 – The meaning of the term’s reliability, construct and concurrent validity could be briefly explained in the introduction, as may be reader dealing with questionnaires validation for the first time.

Reply: We have modified the corresponding paragraph as follow: “Thus, this study aimed to determine the test-retest reliability indicating the stability of the FFQ over time through repeated measures at two different time points. Furthermore, we determined concurrent or relative validity which indicates the amount of agreement between two different measures, and construct validity a measure of the concept that it’s intended to measure. For this purpose, we compared dietary data obtained by the FFQ with those derived by food records and biological markers of food intake. Additionally, we analyzed these validity domains along with changes in adherence to the MedDiet.” (lines: 91-98)

Line 89 – The author mention “we analyzed these validity domains along with changes in adherence to the MedDiet”. Why? Why specifically the MedDiet?

Reply: The reviewer is right that there are numerous healthy a priori dietary patterns. These women are living in a Mediterranean country and, therefore, it is of more interest to determine the validity of the questionnaire concerning the adherence to this dietary pattern than for example the adherence to the DASH diet.    

Line 135 – Minimum validation parameters of the analytical methods should be presented, namely the limits of quantification.

Reply: This has been done. We have included the following information in the method section of the manuscript: “The limits of quantification for biological markers were 9.86 to 32.89 ng/ml, 10.0 to 16.8 ng/ml, equal or less than 90 pg/ml and equal or less than 0,70 ng/ml for hydroxytyrosol, fatty acids, vitamin B12 and folic acid, respectively”.  (lines: 215-217)

Line 157 – Where smoking pregnant women included in the analysis?

Reply: Yes, we have included smoking pregnant women. There were only 4 women who reported to smoke.

Exclusion criteria of participants are not described.

Reply: We have added the following information in the method section of the manuscript:” Exclusion criteria were any of the following: fetal anomalies including chromosomal abnormalities, structural malformations or congenital infections detected prenatally; neonatal abnormalities diagnosed after birth; no possibility to perform additional visits; participation in another trial; and maternal mental retardation or other mental or psychiatric disorders”. (lines: 137-141)

Line 166 – The authors should include the methodology used to control covariables.

Tables 1 and 6 – In the first 3 lines there is no indication that mean values are being presented, only SD.

Reply: Age, alcohol consumption, and smoking were not used as co-variables in statistical models. We presented these variables in Table 1 as an information for the reader about the main characteristics of the present population. We have renamed “co-variables” “other variables” (line:  193)

We have indicated that: “Mean with standard deviation (SD) and proportions (%) were calculated for the characteristics of the subjects (shown in Table 1)” (lines: 231-232)

Tables 2, 3 and 4 – p values of Interclass or Spearman correlations should be presented, and more considered while discussing the results. Why only Table 5?

Reply: All correlations in table 2 were significant. This is stated in the footnote of the table 2. We have included this information in tables 3 and 4.

Furthermore, significance of correlations has been added to the discussion.

“We found a significant test-retest reliability for each of the 12 food groups and 17 nutrients. (lines: 279,280)

“All correlations were significant with the exception for olive oil and vitamin B12 for non-standardized  and olive oil, vitamin B12, and monounsaturated fat for standardized foods and nutrients (Tables 3 and 4). The degree of correlations ranged from poor (olive oil ρ = 0.12) to good (dairy products ρ = 0.63) for foods and from poor (vitamin B12 ρ = 0.10) to moderate (vitamin C ρ = 0.47) for nutrients.”  (lines: 331-336 )

Line 281 – The sentence “The MDS significantly (p <0.001) overestimated the ratings (by 12%) compared to the corresponding score obtained by the FFQ” is not clear.

 Reply: We have rephrased the sentence for better clarity as follows: “The FFQ  significantly (p<0.001) overestimated the MedDiet score (by 12%) compared to the corresponding MedDiet score derived by the reference method”.  (lines:  391-393)

Reviewer 2 Report

Juton et al. have presented a reliability and validation analysis of a food frequency questionnaire specifically for pregnant women at high risk to develop fetal growth restrictions.

Whilst this is a well thought out and written paper, there are a number of additions which could be implemented to make it of a higher standard.

Introduction –

The introduction states food records and 24 hour recalls are ‘gold standards’, however specifically weighed food records are the gold standard and provide more accurate data than 24 hour recalls.

From my understanding, the participants are entering an intervention study during the delay between the first and second administration of the FFQs. Whilst 2 of the 3 intervention arms are unlikely to cause changes in the diets consumed, the nutrition programme based on MedDiet is likely to have caused participants to change their diet. This needs to be highlighted in the discussion and comments made on the implications of this to the results seen.

Methods –

Further details on the FFQ are needed – over what time period does the FFQ collect data, previous 12 months? Is the FFQ a paper form or an online form? What modifications have been made to the FFQ and were these modifications after the published validation study mentioned?

Further details on the food records are required – did they recall their diet over the past 7 days at the meeting or were participants writing down their food diary on a daily basis during that 7-day period? Did they also include estimates of grams/cups consumed of foods or were standard serving sizes used? Were recipes provided for homemade dishes to be used in nutrient intake analysis? The methods state the food records were collected at follow-up visits, but was it just one follow up visit and not multiple (again phrasing suggesting multiple follow up visits is used in Line 158)?

Can it be clearly stated that the MDS was calculated at both time points, and that biological markers were only analysed at the baseline time point (if this is the case).

Both MedDiet and MDS are used as abbreviations, can just one be used consistently throughout the manuscript.

Statistical Analysis –

It needs to be made clear data from baseline or follow up is being used, i.e. from Line 171 I assume it is only baseline data being used from them onwards?

Were the nutrients/food groups normally distributed (what test stat was used to determine) or did some require log transformation? This section states Pearson’s correlation coefficients were used which are less ideal to use if the data is not normally distributed, however Tables 3 and 4 state Spearman’s correlation coefficients were used which are more suitable for non-normally distributed data.

Were any data removed due to extreme energy intakes or they were all within a normal gender specific range?

The statistical analysis section does not mention analysis including the biomarkers.

Given there are three measurements (food frequency questionnaire, reference method/food recall and biomarker) available, a method of triads analysis would improve the novel aspect of the manuscript.

Results –

Can the MDS scores at both baseline and follow up being included in the first paragraph of the results section.

Can energy intake be included in Table 1.

Table 1 includes education and current smokers however these are not used in the analysis or mentioned elsewhere in the paper so should be removed if not going to be further discussed in conjunction with the results.

Tables 3 and 4 would benefit from an extra column stating the level of agreement e.g. moderate, fair etc. so it is easily identifiable for the reader.

Only nut intake was analysed with α-linolenic acid biological marker, however there are other dietary sources of this including tofu, canola oil, and pumpkin seeds. Were these items included in the FFQ or can it be stated that they were not included and the only source of α-linolenic acid in the FFQ was nuts. For hydroxytyrosol, olives are the other dietary source, again were olives included in the FFQ and if not, can it be stated that they were not included and the only source of hydroxytyrosol in the FFQ was olive oil.

Can lines 281 – 286 be checked, the MDS derived by FFQ was higher than the MDS derived by food recalls, whereas these lines stated the MDS overestimated ratings compared to the score obtained by the FFQ. Furthermore, it states it was significantly overestimated, then that they were not significantly different, and then that they were correlated, and needs some editing to be explicit.

Lines 291 states 15 nutrients and then 17 nutrients were evaluation, why is there a discrepancy?

Discussion –

A section on the limitations of the food recall method needs to be included and that weighed food records would have given more accurate data. A section on the sources of error of the FFQ also needs to be included – what foods may have been missing from the FFQ, were there restrictions imposed by a fixed list of foods, errors in memory of foods consumed over the year, assumption of average serving sizes for analysis?

There is a lack of comment on why the nutrients/food groups with poor agreement had this observed when others had better agreement. What might the reasons be for this, potentially agreement may be affected by reporting of items in the FFQ that are not consumed during the 3-day food recall and vice versa. Other reasons that could potentially explain why the FFQ is not measuring these groups as accurately could be stated.

Given that several foods and nutrients found poor agreement, a recommendation that the FFQ should be used selectively if measuring absolute intakes in these items should be included in the conclusions.

Line 388 states 17 nutrients were significantly associated, however in supplementary table 1 there are only 8 of the 17 nutrients significantly associated.

Author Response

Whilst this is a well thought out and written paper, there are a number of additions which could be implemented to make it of a higher standard.

Reply: Thank you. We have addressed the reviewer’s concerns point by point.

The introduction states food records and 24-hour recalls are ‘gold standards’, however specifically weighed food records are the gold standard and provide more accurate data than 24 hour recalls.

Reply: The reviewer is correct. We have modified the corresponding paragraph accordingly: “The accuracy of an FFQ can be determined by comparing food and nutrient intake information collected by this instrument with those obtained using a reference method. Weighed food records are considered the gold standard method; however, the application of this method is often not feasible. Therefore, food records and 24hR are widely used as the reference method. But these instruments are not free from measurement errors caused by memory bias among others [4]. These markers are in contrast to auto reported data objective measures of food intake. It is, therefore, appropriate to include additional biological markers of food intake in the evaluation of the validity of an FFQ.” (lines: 75-81)

From my understanding, the participants are entering an intervention study during the delay between the first and second administration of the FFQs. Whilst 2 of the 3 intervention arms are unlikely to cause changes in the diets consumed, the nutrition programme based on MedDiet is likely to have caused participants to change their diet. This needs to be highlighted in the discussion and comments made on the implications of this to the results seen.

Reply: Thank you for this comment. The reviewer is correct. Dietary counseling during pregnancy in one third of the participants has very likely changed dietary habits of these pregnant women. We have included the following text:

“Additionally, the test-retest reliability of diet in these studies is based on the assumption that differences found between the two estimations are mainly due to measurement errors and less to alterations in dietary habits. In the present study one third of the pregnant women were allocated to a nutritional intervention program and have, therefore, changed their dietary habits during the test-retest period. This fact might partially explain the magnitude of the test-retest reliability of the present FFQ.”  (lines:    )

Methods –

Further details on the FFQ are needed – over what time period does the FFQ collect data, previous 12 months? Is the FFQ a paper form or an online form? What modifications have been made to the FFQ and were these modifications after the published validation study mentioned?

Reply:  Thank you for your comments. Regarding your questions:

  • The FFQ collected data on the previous 12 months. We asked for the dietary intake of the last year, in order to obtain nutritional information about habitual intake. We added this information in line 111 to 112.
  • Our FFQ is an optically-readable paper form. In this case, trained interviewer collected nutritional data with the FFQ. We added this information in line 113.
  • The modifications of the FFQ published by Fernández-Ballart are mainly due to the food databases that we used to estimate nutrients intake. In our case, we used CESNID and Moreiras. This information could be found in the section2. Dietary Assessment. Moreover, we added few food-items to our FFQ in order to adapt the habitual food products consumption in our geographical area, such as sweeteners, soy milk and other plant-based milks, avocado, dark chocolate, different snacks and non-alcoholic beverages such as non-alcoholic beer, infusions and water. We added this information in line 120-123.

Further details on the food records are required – did they recall their diet over the past 7 days at the meeting or were participants writing down their food diary on a daily basis during that 7-day period? Did they also include estimates of grams/cups consumed of foods or were standard serving sizes used? Were recipes provided for homemade dishes to be used in nutrient intake analysis? The methods state the food records were collected at follow-up visits, but was it just one follow up visit and not multiple (again phrasing suggesting multiple follow up visits is used in Line 158)?

Reply: We have addressed your comments as follow:

  • The participants filled a 7-days food dairy of the previous 7 days before the meeting. In this sense, they attended the visit with the 7-days food dairy completed and trained interviewers reviewed carefully the content and asked for some doubts or lack of nutritional information such as the use of oil for cooking and dressing, the use of spices such as salt, among others. We clarify this information about the food records in line 165.
  • As we previously mentioned, the participants detailed their dietary intake, including the estimated grams or portions sizes. Moreover, in the food diary is also described the instructions with the portion sizes and how to provide this information in the food dairy. We added this information in line 124 to 128.
  • We used traditional recipes for the nutrient intake analysis. However, it is important to mention that participants provide their usual diet, which information was translated to nutritional data. We added this clarification in Line 173
  • Thank you for your suggestion. Regarding the follow-up visits: this information was collected in 2 visits:
    • Baseline visit (at 19-24 weeks of gestation)
    • Final visit (at 34-36 weeks of gestation)

We added this information about the visits in line 151

Table 1 includes education and current smokers however these are not used in the analysis or mentioned elsewhere in the paper so should be removed if not going to be further discussed in conjunction with the results.

Reply:  The reviewer is right that the variables age, alcohol consumption, and smoking were not used as co-variables in statistical models. We presented these variables in Table 1 as an information for the reader about the main characteristics of the present population. We have renamed “co-variables” “other variables” (line: 220 ).

Can it be clearly stated that the MDS was calculated at both time points, and that biological markers were only analysed at the baseline time point (if this is the case). Reply: This has been done: “The MedDiet score was calculated according to Trichopoulou and colleagues’ method at baseline and follow-up [11]. The MedDiet score calculated at follow-up was used to assess concurrent validity. Additionally, biological markers analyzed at follow-up were used for the correlation with the corresponding food intake at follow-up”.  (lines: 177-180)

Both MedDiet and MDS are used as abbreviations, can just one be used consistently throughout the manuscript.

Reply: This has been done.  

Statistical Analysis –

It needs to be made clear data from baseline or follow up is being used, i.e. from Line 171 I assume it is only baseline data being used from them onwards?

Reply:  For the analysis of concurrent (relative) and construct validity we used data from the FFQ administer at follow-up and data from food records (reference method). We have clarified this in the statistic section of the manuscript. (lines:234-236)

Were the nutrients/food groups normally distributed (what test stat was used to determine) or did some require log transformation? This section states Pearson’s correlation coefficients were used which are less ideal to use if the data is not normally distributed, however Tables 3 and 4 state Spearman’s correlation coefficients were used which are more suitable for non-normally distributed data.

Reply: We apologize this error in the statistic section. We didn’t log transform variables when it was not strictly necessary as in the case for the ICC to avoid deletion of 0 values. Therefore, we presented Spearman coefficients in tables 3 and 4. We have corrected the statistic section accordingly: “The relative validity of the FFQ (test method) against the food records (reference method) was first assessed by calculating the Spearman correlation coefficient.” (lines: 236,237)

Were any data removed due to extreme energy intakes or they were all within a normal gender specific range?

Reply: There were no value below 1300kcal/d and only 3 cases reported an intake of more than 3500 kcal/d (3508, 3749, and 3872). After careful checking of those cases we have decided not to exclude them from analysis.

The statistical analysis section does not mention analysis including the biomarkers.

Reply: We regret this failure. We have added the following paragraph to the statistic section: “Additionally, we determined Pearson correlation coefficients between biological markers and the corresponding food and nutrient intake.”  (lines:237-246))

Given there are three measurements (food frequency questionnaire, reference method/food recall and biomarker) available, a method of triads analysis would improve the novel aspect of the manuscript.

Reply: We have addressed several dimensions of the validity of the questionnaire. The data are presented in 6 tables and 1 Figure. This gives the reader an extensive information on the utility of this new questionnaire for pregnant women.  We do not believe that an additional analysis on triads will give a substantial better insight in the validity of the questionnaire.

Results –

Can the MDS scores at both baseline and follow up being included in the first paragraph of the results section.

Reply: This has been done: “The mean of the MedDiet score at baseline and follow up was 4.0±1.5 and 4.1±1.6, respectively”. (lines: 277,278)

Can energy intake be included in Table 1.

Reply: This has been done. 

Table 1 includes education and current smokers however these are not used in the analysis or mentioned elsewhere in the paper so should be removed if not going to be further discussed in conjunction with the results.

Reply:  The reviewer is right that the variables age, alcohol consumption, and smoking were not used as co-variables in statistical models. We presented these variables in Table 1 as an information for the reader about the main characteristics of the present population. We have renamed “co-variables” “other variables” (line: 220).

Tables 3 and 4 would benefit from an extra column stating the level of agreement e.g. moderate, fair etc. so it is easily identifiable for the reader.

Reply: This has been done.

Only nut intake was analysed with α-linolenic acid biological marker, however there are other dietary sources of this including tofu, canola oil, and pumpkin seeds. Were these items included in the FFQ or can it be stated that they were not included and the only source of α-linolenic acid in the FFQ was nuts. For hydroxytyrosol, olives are the other dietary source, again were olives included in the FFQ and if not, can it be stated that they were not included and the only source of hydroxytyrosol in the FFQ was olive oil.

Reply: We have included the requested information in the result section:   The comparison of the biomarker α-linolenic acid was limited to nut consumption because there are no other relevant food sources α-linolenic acid included in the FFQ. Hydroxytyrosol was correlated with olive oil and olives, both principal sources of this bioactive compound” (lines: 339-343)

We have included the correlation between hydroxytyrosol and olives in Table 5.

Can lines 281 – 286 be checked, the MDS derived by FFQ was higher than the MDS derived by food recalls, whereas these lines stated the MDS overestimated ratings compared to the score obtained by the FFQ. Furthermore, it states it was significantly overestimated, then that they were not significantly different, and then that they were correlated, and needs some editing to be explicit.

Reply: We have modified this paragraph as follows: “The mean ratings of the MedDiet score derived by the FFQ at baseline and follow-up 4.0 ± 1.5 and 4.1 ± 1.6, respectively, and for dietary records 4.0 ± 1.5. The FFQ significantly (p<0.001) overestimated the MedDiet score (by 12%) compared to the corresponding MedDiet score derived by the reference method. However, no proportional bias was found (β coefficient 0.072; 95% CI −0.064, 0.204; p <0.287) (Table 6 and Figure 1) across score ratings. The Pearson coefficient revealed a moderate and significant correlation (0.46 and p <0.001) between the scores derived by the dietary records and FFQ.” (lines: 390-396)

Lines 291 states 15 nutrients and then 17 nutrients were evaluation, why is there a discrepancy? We apologize this error. 17 nutrients were evaluated. We have corrected this error accordingly.

Discussion –

A section on the limitations of the food recall method needs to be included and that weighed food records would have given more accurate data. A section on the sources of error of the FFQ also needs to be included – what foods may have been missing from the FFQ, were there restrictions imposed by a fixed list of foods, errors in memory of foods consumed over the year, assumption of average serving sizes for analysis?

Reply: We have included the requested information in the limitation section:

Finally, as in all validation studies, an inherent limitation is that reference methods such as multiple dietary recalls or records are themselves not free from error [16]. Food records for example requires motivated subjects and place a high burden on the participants. The ideal choice of the reference method is weighed food records, which is considered the “gold standard. However, the administration of food records or weighed food records may lead to participants changing their diet during a recording period. Food frequency questionnaires on the other hand are prone to memory bias because these questionnaires asked for the retrospective food intake. Furthermore, average consumption frequency of seasonal foods is especially critical and the fixed food list in fixed portion sizes are other sources of measurement error.”  (lines: 546-555)

There is a lack of comment on why the nutrients/food groups with poor agreement had this observed when others had better agreement. What might the reasons be for this, potentially agreement may be affected by reporting of items in the FFQ that are not consumed during the 3-day food recall and vice versa. Other reasons that could potentially explain why the FFQ is not measuring these groups as accurately could be stated.

Reply: We have included the following paragraph to the discussion:  “The poor concurrent validity of olive oil in the present study was somewhat surprising because it is a characteristic food of the Mediterranean diet and hence it is unlikely that the relatively short reporting time of the reference method was a reason for this finding. However, the objective measure of olive oil consumption by its corresponding biological marker revealed a better correlation. In contrast, a poor correlation of vitamin B12 derived by the FFQ with the corresponding data from the reference method and the biological marker were found for both. One might argue that this possibly reflects the difficulty in estimating meat servings but the correlations of meat and processed meat between methods were substantially better than that for vitamin B12. The poor correlation of vitamin B12 derived by the FFQ with its biological marker might be biased by dietary supplements containing vitamin B12.”  (lines: 487-497)

Given that several foods and nutrients found poor agreement, a recommendation that the FFQ should be used selectively if measuring absolute intakes in these items should be included in the conclusions.

Reply: The use of FFQs for the assessment of absolute food intake is limited. We have addressed  this topic in the limitation section: “Finally, the use of the FFQ to present data of absolute intakes of foods and nutrients is limited without a prior calibration of these data by a reference method. This is especially the case for foods and nutrients with poor concurrent validity and concordance.”  (lines:  555-558)

Line 388 states 17 nutrients were significantly associated, however in supplementary table 1 there are only 8 of the 17 nutrients significantly associated.

Reply: The reviewer is correct. We have modified the sentence to read: “We found that intakes of 17 nutrients were associated in the anticipated direction with MedDiet score ratings derived by the FFQ., The associations were significant for nearly 50% of the nutrients.”

(lines: 402-404).

Round 2

Reviewer 2 Report

All comments have been sufficiently responded to and edits are satisfactory.

Unfortunately with some participants being included in an intervention which changed their diet throughout the study, I feel this cohort was not the most suitable for a reliability study of a food frequency questionnaire. This issue will affect the results, and therefore the impact of the paper within the extant literature, and future use as a representation of proof of reliability of the food frequency questionnaire.